# The Limit Points of (Optimistic) Gradient Descent in Min-Max Optimization

**Constantinos Daskalakis**
CSAIL
MIT
Cambridge, MA 02138
costis@csail.mit.edu

**Ioannis Panageas**
ISTD
SUTD
Singapore, 487371
ioannis@sutd.edu.sg

## Abstract

Motivated by applications in Optimization, Game Theory, and the training of Generative Adversarial Networks, the convergence properties of first order methods in min-max problems have received extensive study. It has been recognized that they may cycle, and there is no good understanding of their limit points when they do not. When they converge, do they converge to local min-max solutions? We characterize the limit points of two basic first order methods, namely Gradient Descent/Ascent (GDA) and Optimistic Gradient Descent Ascent (OGDA). We show that both dynamics avoid unstable critical points for almost all initializations. Moreover, for small step sizes and under mild assumptions, the set of OGDA-stable critical points is a superset of GDA-stable critical points, which is a superset of local min-max solutions (strict in some cases). The connecting thread is that the behavior of these dynamics can be studied from a dynamical systems perspective.

## 1 Introduction

The celebrated min-max theorem was a founding stone in the development of Game Theory [21], and is intimately related to strong linear programming duality [1], Blackwell's approachability theory [3], and the theory of no-regret learning [5]. The theorem states that if $f(\mathbf{x}, \mathbf{y})$ is a convex-concave function, and $\mathcal{X}, \mathcal{Y}$ are compact and concave subsets of Euclidean space, then

$$\min_{x \in \mathcal{X}} \max_{y \in \mathcal{Y}} f(\mathbf{x}, \mathbf{y}) = \max_{y \in \mathcal{Y}} \min_{x \in \mathcal{X}} f(\mathbf{x}, \mathbf{y}). \tag{1}$$

If $f(\mathbf{x}, \mathbf{y})$ represents the payment of the $\mathcal{X}$ player to the $\mathcal{Y}$ player under choices of strategies $\mathbf{x} \in \mathcal{X}$ and $\mathbf{y} \in \mathcal{Y}$ by these two players, the min-max theorem reassures us that an equilibrium of the game exists, and that the equilibrium payoffs to both players are unique.

What does not follow directly from the min-max theorem is whether there exist dynamics via which players would arrive at equilibrium if they were to follow some simple rule to update their current strategies. This has been the topic of a long line of investigation starting with Julia Robinson's celebrated analysis of fictitious play [4, 20], and leading to the development of no-regret learning [5].

Renewed interest in this problem has been recently motivated by the task of training Generative Adversarial Networks (GANs) [9, 2], where two deep neural networks, the generator and the discriminator, are trained in tandem using first order methods, aiming at solving a min-max problem, of the following form, albeit typically with a non convex-concave objective function $f(\mathbf{x}, \mathbf{y})$:

$$\inf_{x \in \mathcal{X}} \sup_{y \in \mathcal{Y}} f(\mathbf{x}, \mathbf{y}). \tag{2}$$

Here $\mathbf{x}$ represents the parameters of the generator deep neural net, $\mathbf{y}$ represents the parameters of the discriminator neural net, and $f(\mathbf{x}, \mathbf{y})$ is some measure of how close the distribution generated by the generator appears to the true distribution from the perspective of the discriminator.

Min-max optimization in non convex-concave settings is a central problem for many research communities, however our knowledge is very limited from optimization perspective. Moreover, for such applications of first-order methods to min-max problems in Machine Learning, it is especially important that the last-iterate maintained by the min and the max dynamics converges to a desirable solution. Unfortunately, even when $f(\mathbf{x}, \mathbf{y})$ is convex-concave, it is rare that guarantees are known for the last iterate (see [17, 15, 13] for continuous time learning dynamics that may cycle). Some guarantees are known for continuous-time dynamics [6], but for discrete-time dynamics it is typically only shown that the average-iterates converge to min-max equilibrium. Recent work of [7] shows that, while Gradient Descent/Ascent (GDA) dynamics performed by the min/max players may diverge, the Optimistic version dynamics of [18] exhibit last iterate convergence to min-max solutions (which we shall call Optimistic Gradient Descent/Ascent (OGDA)), whenever $f(\mathbf{x}, \mathbf{y})$ is linear in $\mathbf{x}$ and $\mathbf{y}$[1]. The goal of our paper is to understand the limit points of GDA and OGDA dynamics (points that last iterate might converge to) for general functions $f(\mathbf{x}, \mathbf{y})$. In particular, we answer the following questions:

- are the stable limit points of GDA and OGDA locally min-max solutions?
- how do the stable limit points of GDA and OGDA relate to each other?

We provide answers to these questions after defining our dynamics of interest formally.

**GDA and OGDA Dynamics.**    Assume from now on that $\mathcal{X} = \mathbb{R}^n$, $\mathcal{Y} = \mathbb{R}^m$ and $f$ is a real-valued function in $C^2$, the space of twice-continuously differentiable functions (unconstrained case). Perhaps the most natural approach to solve (2) is by doing gradient descent on $\mathbf{x}$ and gradient ascent on $\mathbf{y}$ (GDA), i.e.,

$$
\begin{aligned}
\mathbf{x}_{t+1} &= \mathbf{x}_t - \alpha \nabla_{\mathbf{x}} f(\mathbf{x}_t, \mathbf{y}_t), \\
\mathbf{y}_{t+1} &= \mathbf{y}_t + \alpha \nabla_{\mathbf{y}} f(\mathbf{x}_t, \mathbf{y}_t),
\end{aligned}
\tag{3}
$$

with some constant step size $\alpha > 0$[2]. However, there are examples (functions $f$ and initial points $(\mathbf{x}_0, \mathbf{y}_0)$) in which the system of equations (3) cycles (see [7]). To break down this behavior, the authors in [7] analyzed another optimization algorithm which is called Optimistic Gradient Descent/Ascent (OGDA)[3], the equations of which boil down to the following:

$$
\begin{aligned}
\mathbf{x}_{t+1} &= \mathbf{x}_t - 2\alpha \nabla_{\mathbf{x}} f(\mathbf{x}_t, \mathbf{y}_t) + \alpha \nabla_{\mathbf{x}} f(\mathbf{x}_{t-1}, \mathbf{y}_{t-1}), \\
\mathbf{y}_{t+1} &= \mathbf{y}_t + 2\alpha \nabla_{\mathbf{y}} f(\mathbf{x}_t, \mathbf{y}_t) - \alpha \nabla_{\mathbf{y}} f(\mathbf{x}_{t-1}, \mathbf{y}_{t-1}).
\end{aligned}
\tag{4}
$$

One of their key results was to show convergence to the $\min \max$ solution for the case of bilinear objective functions, namely $f(\mathbf{x}, \mathbf{y}) = \mathbf{x}^\top A \mathbf{y}$.

**Our contribution and techniques:** In this paper we analyze Gradient Descent/Ascent (GDA) and Optimistic Gradient Descent/Ascent (OGDA) dynamics applied to min-max optimization problems. Our starting point is to show that both dynamics avoid their unstable fixed points (GDA-unstable and OGDA-unstable respectively, as defined in Section 1.1). This is shown by using techniques from dynamical systems, following the line of work of recent papers in Optimization and Machine Learning [14, 11, 10]. In a nutshell we show that the update rules of both dynamics are local diffeomorphisms[4] and we then make use of Center-Stable manifold theorem A.1 (see supplementary material). These results are given as Theorems 2.2, 3.2. We note that it is very crucial to show that the update rule of the dynamics is a diffeomorphism, otherwise avoiding "linearly" unstable points cannot be shown globally. One important step in our approach is the construction of a dynamical system for OGDA in order to apply dynamical systems' techniques.

We next study the set of stable fixed points of GDA dynamics and their relation to locally min-max solutions, called *local min-max*.[5]). Informally, a local min-max critical point $(\mathbf{x}^*, \mathbf{y}^*)$ satisfies the following: compared to value $f(\mathbf{x}^*, \mathbf{y}^*)$, if we fix $\mathbf{x}^*$ and perturb $\mathbf{y}^*$ infinitesimally, the value of $f$

does not *increase* and similarly if we fix $\mathbf{y}^*$ and perturb $\mathbf{x}^*$ infinitesimally, the value of $f$ does not *decrease*. We show that the set of stable fixed points of GDA is a superset of the set of local min-max and there are functions in which this inclusion is strict. This is given in Lemmas 2.4, 2.7, 2.5.

Finally, we analyze OGDA dynamics which is a bit trickier than GDA due to the nature of the dynamics, namely the existence of memory in the dynamics: the next iterate depends on the gradient of the current and previous point. We construct a dynamical system that captures OGDA dynamics (see Equation (7)), using a construction that is commonly employed in differential equations. Importantly, we establish a mapping (relation) between the eigenvalues of the Jacobian of the update rules of both GDA and OGDA, showing that OGDA stable fixed points is a superset of GDA stable ones (under mild assumptions on the stepsize), namely (we suggest the reader to see first Remark 1.5 to avoid confusion):

$$\text{Local min-max} \quad \subset \quad \text{GDA-stable} \quad \subset \quad \text{OGDA-stable}$$

We note that the inclusion above are strict.

**Notation:** Vectors in $\mathbb{R}^n, \mathbb{R}^m$ are denoted in boldface $\mathbf{x}, \mathbf{y}$. Time indices are denoted by subscripts. Thus, a time indexed vector $\mathbf{x}$ at time $t$ is denoted as $\mathbf{x}_t$. We denote by $\nabla_{\mathbf{x}} f(\mathbf{x}, \mathbf{y})$ the gradient of $f$ with respect to variables in $\mathbf{x}$ (of dimension the same as $\mathbf{x}$) and by $\nabla^2_{\mathbf{xy}} f$ the part of the Hessian in which the derivative of $f$ is taken with respect to a variable in $\mathbf{x}$ and then a variable in $\mathbf{y}$. We use the letter $J$ to denote the Jacobian of a function (with appropriate subscript), $\mathbf{I}_k, \mathbf{0}_{k \times l}$ to denote the identity and zero matrix of sizes $k \times k$ and $k \times l$ respectively[6], $\rho(A)$ for the spectral radius of matrix $A$ and finally we use $f^t$ to denote the composition of $f$ by itself $t$ times.

Finally, we would like to note that all the missing proofs can be found in the supplementary material.

## 1.1 Important Definitions

We have already stated our min-max problem of interest (2) as well as the Gradient Descent/Ascent (GDA) dynamics (3) and Optimistic Gradient Descent/Ascent (OGDA) dynamics (4) that we plan to analyze. We provide some further definitions.

**Dynamical Systems.** A recurrence relation of the form $\mathbf{x}_{t+1} = w(\mathbf{x}_t)$ is a discrete time dynamical system, with update rule $w : \mathcal{S} \to \mathcal{S}$ for some convex set $\mathcal{S} \subset \mathbb{R}^n$. Function $w$ is assumed to be continuously differentiable for the purpose of this paper. The point $\mathbf{z}$ is called a *fixed point* or *equilibrium* of $w$ if $w(\mathbf{z}) = \mathbf{z}$. We will be interested in the following standard notions of fixed point stability.

**Definition 1.1** ((Linear) stability)**.** *Let $w$ be continuously differentiable. We call a fixed point $\mathbf{z}$* linearly stable *or just* stable *if, for the Jacobian $J$ of $w$ computed at $\mathbf{z}$, it holds that its spectral radius $\rho(J)$ is at most one and otherwise we call it* linearly unstable *or just* unstable.

**Definition 1.2** (Lyapunov and Asymptotic Stability)**.** *A fixed point $\mathbf{z}$ of $w$ is called* Lyapunov stable *if, for every $\epsilon > 0$, there exists a $\delta = \delta(\epsilon) > 0$ such that if $\mathbf{x} \in \mathcal{B}_\delta$ with $\mathcal{B}_\delta = \{\mathbf{y} \in \mathcal{S} : \|\mathbf{y} - \mathbf{z}\| < \delta\}$[7] we have that $\|w^n(\mathbf{x}) - \mathbf{z}\| < \epsilon$ for every $n \geq 0$. That is, if dynamics starts close enough to $\mathbf{z}$, it remains close for all times.*

*A fixed point $\mathbf{z}$ of $w$ is called (locally) asymptotically stable (or attracting) if it is Lyapunov stable and there exists a $\delta > 0$ such that, for all $\mathbf{x} \in \mathcal{B}_\delta$ we have that $\|w^n(\mathbf{x}) - \mathbf{z}\| \to 0$ as $n \to \infty$. That is, there is a small neighborhood around $\mathbf{z}$ so that, for all initializations in that neighborhood, the dynamics converges to $\mathbf{z}$.*

**Definition 1.3** (Hyperbolicity)**.** *We call a fixed point $\mathbf{z}$ hyperbolic iff the Jacobian $J$ of $w$ computed at $\mathbf{z}$ has no eigenvalues with absolute value $1$.*

The following are well-known facts.

**Proposition 1.4** (e.g. [8])**.** *If the Jacobian of the update rule at a stable fixed point $\mathbf{z}$ has spectral radius less than one, then the fixed point is asymptotically stable. Therefore, if a fixed point $\mathbf{z}$ is hyperbolic, then linear stability implies asymptotic stability.*

**Remark 1.5** (**Fixed points of GDA, OGDA dynamics**). *It is easy to see that a fixed point of the GDA dynamics (3) arises whenever* $(\mathbf{x}_{t+1}, \mathbf{y}_{t+1}) = (\mathbf{x}_t, \mathbf{y}_t)$, *or in other words whenever* $(\mathbf{x}_t, \mathbf{y}_t) = (\mathbf{x}, \mathbf{y})$ *such that* $\nabla f(\mathbf{x}, \mathbf{y}) = \mathbf{0}$.

*Since the OGDA dynamics (4) has memory, it is more appropriate to think of the dynamics as mapping a quadruple* $(\mathbf{x}_t, \mathbf{y}_t, \mathbf{x}_{t-1}, \mathbf{y}_{t-1})$ *to a quadruple* $(\mathbf{x}_{t+1}, \mathbf{y}_{t+1}, \mathbf{x}_t, \mathbf{y}_t)$. *In this case, a fixed point arises whenever* $(\mathbf{x}_{t+1}, \mathbf{y}_{t+1}, \mathbf{x}_t, \mathbf{y}_t) = (\mathbf{x}_t, \mathbf{y}_t, \mathbf{x}_{t-1}, \mathbf{y}_{t-1})$, *or in other words whenever* $(\mathbf{x}_t, \mathbf{y}_t, \mathbf{x}_{t-1}, \mathbf{y}_{t-1}) = (\mathbf{x}, \mathbf{y}, \mathbf{x}, \mathbf{y})$ *and* $\nabla f(\mathbf{x}, \mathbf{y}) = \mathbf{0}$.

*We should stress in particular that whenever we say that the set of OGDA-stable fixed points is a super-set of the GDA-stable fixed points, we will be somewhat abusing notation, since the fixed points of OGDA lie in* $\mathbb{R}^{2n+2m}$ *while the fixed points of GDA lie in* $\mathbb{R}^{n+m}$. *However, as discussed above, a fixed point of OGDA is of the form* $(\mathbf{x}, \mathbf{y}, \mathbf{x}, \mathbf{y})$, *and we can thus project it to its first two components without any loss of information to obtain a point in* $\mathbb{R}^{n+m}$. *When we relate fixed points of OGDA to fixed points of GDA we will implicitly apply this projection.*

Given Proposition 1.4, it follows that spectral analysis of the Jacobian of the fixed points can give us qualitative information about the local behavior of the dynamics. Unless otherwise specified, throughout this paper, whenever we say "stable" we mean linearly stable. GDA/OGDA-stable critical points are critical points that are stable with respect to GDA/OGDA dynamics (for fixed stepsize $\alpha$, otherwise are unstable). Moreover since different choices of stepsize $\alpha$ might give different stability for GDA and OGDA dynamics, we are interested in the case $\alpha$ is "*sufficiently*" small. Therefore in the sections we *characterize* the GDA/OGDA-stable critical points, a point $(\mathbf{x}^*, \mathbf{y}^*)$ is classified as GDA/OGDA-stable if there exists a sufficiently small number $\beta > 0$ such that for all stepsizes $0 < \alpha < \beta$ we have that the $(\mathbf{x}^*, \mathbf{y}^*)$ is a stable fixed point of GDA/OGDA dynamics (in case there exists a small $\beta > 0$ so that for all stepsizes $0 < \alpha < \beta$ we have that $(\mathbf{x}^*, \mathbf{y}^*)$ is an unstable fixed point of GDA/OGDA dynamics, it is classified as GDA/OGDA-unstable).

**Optimization.**    We use the following standard terminology.

**Definition 1.6.** *For a min-max problem* (2) *where* $f$ *is twice continuously differentiable,*

- *A point* $(\mathbf{x}^*, \mathbf{y}^*)$ *is a* critical point *of* $f$ *if* $\nabla f(\mathbf{x}^*, \mathbf{y}^*) = \mathbf{0}$.

- *A critical point* $(\mathbf{x}^*, \mathbf{y}^*)$ *is* isolated *if there is a neighborhood $U$ around* $(\mathbf{x}^*, \mathbf{y}^*)$ *where* $(\mathbf{x}^*, \mathbf{y}^*)$ *is the only critical point.*[8] *Otherwise it is called* non-isolated.

- *A critical point* $(\mathbf{x}^*, \mathbf{y}^*)$ *is a* local min-max point *if there exists a neighborhood $U$ around* $(\mathbf{x}^*, \mathbf{y}^*)$ *so that for all* $(\mathbf{x}, \mathbf{y}) \in U$ *we have that* $f(\mathbf{x}^*, \mathbf{y}) \leq f(\mathbf{x}^*, \mathbf{y}^*) \leq f(\mathbf{x}, \mathbf{y}^*)$.[9]

- *A critical point* $(\mathbf{x}^*, \mathbf{y}^*)$ *is a* strongly local min-max point *if* $\lambda_{\min}(\nabla^2_{\mathbf{xx}} f(\mathbf{x}^*, \mathbf{y}^*)) > 0$ *and* $\lambda_{\max}(\nabla^2_{\mathbf{yy}} f(\mathbf{x}^*, \mathbf{y}^*)) < 0$.

## 1.2   Formal Statement of Results

We present our main results for GDA and OGDA, to be proven in Sections 2 and 3. Some of our claims make use of the following assumptions about the objective function $f$ of (2):

**Assumption 1.7** (Invertibility of Hessian of $f$). $\nabla^2 f$ *(the Hessian of $f$) is invertible for all* $\mathbf{x}, \mathbf{y}$.

**Assumption 1.8** (Non-Imaginary GDA at a Critical Point). *GDA is* non-imaginary *at a critical point* $(x^*, y^*)$ *of* $f$ *iff*

$$H = \begin{pmatrix} -\nabla^2_{\mathbf{xx}} f & -\nabla^2_{\mathbf{xy}} f \\ \nabla^2_{\mathbf{yx}} f & \nabla^2_{\mathbf{yy}} f \end{pmatrix} \tag{5}$$

*has no eigenvalue whose real part is* 0. $H$ *captures the difference* $\frac{1}{\alpha}(J(\mathbf{x}^*, \mathbf{y}^*) - \mathbf{I}_{n+m})$ *where* $J$ *is the Jacobian of GDA dynamics and* $\mathbf{I}_{n+m}$ *the identity matrix.*

**Remark 1.9.** *To illustrate the nature of the above assumptions, we note that Assumption 1.7 is generically true for quadratic functions. Take an arbitrary quadratic function* $f(\mathbf{x}) = \frac{1}{2}\mathbf{x}^\top Q \mathbf{x}$,

*and define $\tilde{f}(\mathbf{x}) = f(\mathbf{x}) + \frac{1}{2}\mathbf{x}^\top A\mathbf{x}$ where $A$ is a matrix with random entries from some continuous distribution (say uniform in $[-\epsilon, \epsilon]$ for $\epsilon$ small enough). It is not hard to see that $\nabla^2 \tilde{f}$ is invertible with probability one. This is intuitively a "hyperbolicity" assumption of the fixed points of the dynamics. We note that we use this assumption for Lemma 3.1 and also to show that OGDA avoids its unstable fixed points. The stability characterizations do not need this assumption. Moreover, we note that Assumption 1.8 is satisfied when critical point $(\mathbf{x}^*, \mathbf{y}^*)$ is strongly local min-max.*

Our two main results are stated as follows:

**Theorem 1.10** (Inclusion)**.** *Assume $f$ is twice differentiable and $\nabla f$ is Lipschitz with constant $L$.*

- *Let $(\mathbf{x}^*, \mathbf{y}^*)$ be a local min max critical point that satisfies Assumption 1.8. For $\alpha > 0$ sufficiently small it holds that $(\mathbf{x}^*, \mathbf{y}^*)$ is GDA-stable fixed point. There is a function with critical point $(\mathbf{x}^*, \mathbf{y}^*)$ which violates Assumption 1.8, $(\mathbf{x}^*, \mathbf{y}^*)$ is local min-max but not GDA-stable for any $0 < \alpha < \frac{1}{L}$ (Lemmas 2.4, 2.7 and 2.6).*

  *Additionally, if $(\mathbf{x}^*, \mathbf{y}^*)$ is a strongly local min max critical point then Assumption 1.8 is satisfied and for $\alpha > 0$ sufficiently small we get $(\mathbf{x}^*, \mathbf{y}^*)$ is GDA-stable (Remark 2.8).*

  *Finally there is a function with a critical point $(\mathbf{x}^*, \mathbf{y}^*)$ which is not local min-max but it is GDA-stable (for sufficiently small $\alpha > 0$, Lemma 2.5).*

- *Let $(\mathbf{x}^*, \mathbf{y}^*)$ be a GDA-stable fixed point. For $0 < \alpha < \frac{1}{2L}$ it holds that $(\mathbf{x}^*, \mathbf{y}^*)$ is OGDA-stable. Moreover the inclusion is strict, i.e., there is a function with critical point $(\mathbf{x}^*, \mathbf{y}^*)$ which is OGDA-stable but not GDA-stable (for small enough $\alpha > 0$, Lemmas 3.4 and 3.5).*

**Theorem 1.11** (Avoid unstable)**.** *Assume $f$ is twice differentiable and $\nabla f$ is Lipschitz with constant $L$. The set of initial vectors $(\mathbf{x}_0, \mathbf{y}_0)$ so that GDA converges to (linearly) GDA-unstable fixed points (critical points) is of measure zero. Under Assumption 1.7, the set of initial vectors $(\mathbf{x}_1, \mathbf{y}_1, \mathbf{x}_0, \mathbf{y}_0)$ so that OGDA converges to (linearly) OGDA-unstable fixed points (critical points) is of measure zero. These statements are captured by Theorems 2.2 and 3.2.*

## 2 Analysis of Gradient Descent/Ascent

In this section we analyze the local behavior (which carries over to a global characterization under Lemma 2.1 and Center-stable manifold theorem A.1) of GDA dynamics (3). In all our statements (theorems, lemmas etc) we work with real-valued function $f$ that is twice differentiable and we also assume $\nabla f$ is Lipschitz with constant $L$ and that the stepsize satisfies $0 < \alpha < \frac{1}{L}$ (unless stated otherwise in the statement of a lemma/theorem).

### 2.1 Analyzing GDA

We need to show the following lemma in order to use the stable manifold theorem (see Theorem A.1).

**Lemma 2.1** (**GDA is a local diffeomorphism**)**.** *Let $f$ be twice differentiable and $\nabla f$ is Lipschitz with constant $L$. Assume that $0 < \alpha < \frac{1}{L}$. The update rule of the GDA dynamics (3) is a local diffeomorphism.*

**Theorem 2.2** (**Measure zero for GDA**)**.** *Let $f$ be twice differentiable and $\nabla f$ is Lipschitz with constant $L$. Assume that $0 < \alpha < \frac{1}{L}$ and let $h$ be the update rule of the GDA dynamics (3), $(\mathbf{x}^*, \mathbf{y}^*)$ be a GDA-unstable critical point and $W_{GDA}(x^*, y^*)$ be its stable set, i.e.,*

$$W_{GDA}(x^*, y^*) = \{(x_0, y_0) : \lim_k h^k(x_0, y_0) = (x^*, y^*)\}.$$

*It holds that $W_{GDA}(x^*, y^*)$ is of Lebesgue measure zero. Moreover if $W_{GDA}$ is union of the stable sets of all GDA-unstable critical points, then $W_{GDA}$ has also measure zero (namely the proof works for non-isolated critical points).*

The following corollary is immediate from Theorem 2.2.

**Corollary 2.3.** *Let $(\mathbf{x}^*, \mathbf{y}^*)$ be GDA-unstable. Assume $\mu$ is a measure of the starting points $(\mathbf{x}_0, \mathbf{y}_0)$ and is absolutely continuous with respect to the Lebesgue measure on $\mathbb{R}^{n+m}$. Then it holds that*

$$Pr[\lim_t (\mathbf{x}_t, \mathbf{y}_t) = (\mathbf{x}^*, \mathbf{y}^*)] = 0.$$

## 2.2 Characterizing GDA-stability

**Lemma 2.4 (Local min-max are GDA-stable).** *Assume that $0 < \alpha < \frac{1}{L}$ and let $(\mathbf{x}^*, \mathbf{y}^*)$ be a local min-max critical point of $f$ and matrix $H$ (see equations (5)) computed at $(\mathbf{x}^*, \mathbf{y}^*)$ has real eigenvalues. It holds that $(\mathbf{x}^*, \mathbf{y}^*)$ is GDA-stable.*

**Lemma 2.5.** *The converse of Lemma 2.4 is false. There are functions with critical points that are GDA-stable but not local min-max. An example is $f(x,y) = -\frac{1}{8}x^2 - \frac{1}{2}y^2 + \frac{6}{10}xy$[10].*

*Proof.* We provide an example with two variables (so that we can also give a figure). Let $f(x,y) = -\frac{1}{8}x^2 - \frac{1}{2}y^2 + \frac{6}{10}xy$. Computing the Jacobian of the update rule of dynamics (3) at point $(0,0)$ we get that

$$J_{\text{GDA}} = \begin{pmatrix} 1 + \frac{1}{4}\alpha & -\frac{6}{10}\alpha \\ \frac{6}{10}\alpha & 1 - \alpha \end{pmatrix}, \tag{6}$$

Both eigenvalues of $J_{\text{GDA}}$ have magnitude less than 1 (for any $0 < \alpha < \frac{1}{L}$ where $L \leq 1.34$). Finally matrix $H_{\text{GDA}}$ has real eigenvalues. Therefore there exists a neighborhood $U$ of $(0,0)$ so that for all $(x_0, y_0) \in U$, we get that $\lim_t(x_t, y_t) = (0,0)$ for GDA dynamics (3). However it is clear that $(0,0)$ is not a local min-max. See also Figure 1 for a pictorial illustration of the result. □

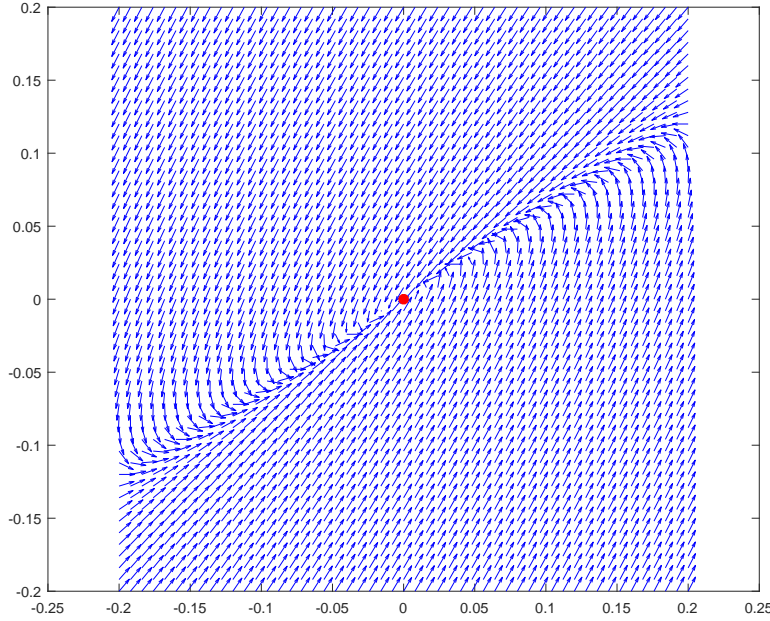

Figure 1: Function $f(x,y) = -\frac{1}{8}x^2 - \frac{1}{2}y^2 + \frac{6}{10}xy$ and $\alpha = 0.001$. The arrows point towards the next step of the Gradient Descent/Ascent dynamics. We can see that the system converges to $(0,0)$ point (GDA-stable), which is not a local min-max critical point.

We end Section 2 by characterizing the case in which $H$ has complex eigenvalues.

**Lemma 2.6 (Imaginary eigenvalues).** *There are functions with critical points that are not GDA-stable but are local min-max when matrix $H$ (see equations (5)) has imaginary eigenvalues.*

We complete the characterization for the relation between GDA-stable critical points and local min-max with the following lemma:

**Lemma 2.7 (Real part nonzero).** *Let $(\mathbf{x}^*, \mathbf{y}^*)$ be a local min-max critical point of $f$ and matrix $H$ (see equations (5)) computed at $(\mathbf{x}^*, \mathbf{y}^*)$ has all its eigenvalues with real part nonzero (i.e., Assumption 1.8). There is a small enough step-size $\alpha > 0$ so that $(\mathbf{x}^*, \mathbf{y}^*)$ is GDA-stable.*

**Remark 2.8.** *If the critical point* $(\mathbf{x}^*, \mathbf{y}^*)$ *is strongly local min-max then* $\lambda_{\max}(H) < 0$ *and hence* $(\mathbf{x}^*, \mathbf{y}^*)$ *is attracting under GDA dynamics, i.e., it holds that Strongly Local min-max* $\subset$ *GDA-stable.*

# 3   Optimistic Gradient Descent/Ascent

The results of the previous section cannot carry over to Optimistic Gradient Descent/Ascent due to the fact that the dynamics has memory and is more challenging to analyze. Here we show that Optimistic Gradient Descent/Ascent avoid OGDA-unstable critical points and we also relate the eigenvalues of the Jacobian of OGDA to the eigenvalues of the Jacobian of GDA. In particular we show that GDA-stable $\subset$ OGDA-stable (inclusion strict). In the beginning we will construct a dynamical system that captures the dynamics of OGDA (4).

## 3.1   Constructing the Dynamical System

We define the function $F$ to be $F(\mathbf{x}, \mathbf{y}, \mathbf{z}, \mathbf{w}) = f(\mathbf{x}, \mathbf{y})$ for all $(\mathbf{x}, \mathbf{y}, \mathbf{z}, \mathbf{w}) \in \mathcal{X} \times \mathcal{Y} \times \mathcal{X} \times \mathcal{Y}$ (think of the last two vector components as dummy for function $F$, its value does not depend on them). Hence it is clear that $\nabla_{\mathbf{z}} F(\mathbf{x}, \mathbf{y}, \mathbf{z}, \mathbf{w}) = \mathbf{0}$ and $\nabla_{\mathbf{w}} F(\mathbf{x}, \mathbf{y}, \mathbf{z}, \mathbf{w}) = \mathbf{0}$. The same holds for $\nabla_{\mathbf{x}} F(\mathbf{z}, \mathbf{w}, \mathbf{x}, \mathbf{y}) = \mathbf{0}$ and $\nabla_{\mathbf{y}} F(\mathbf{z}, \mathbf{w}, \mathbf{x}, \mathbf{y}) = \mathbf{0}$.

We define the following function $g$ which consists of 4 components:

$$
\begin{aligned}
g(\mathbf{x}, \mathbf{y}, \mathbf{z}, \mathbf{w}) &:= (g_1(\mathbf{x}, \mathbf{y}, \mathbf{z}, \mathbf{w}), g_2(\mathbf{x}, \mathbf{y}, \mathbf{z}, \mathbf{w}), g_3(\mathbf{x}, \mathbf{y}, \mathbf{z}, \mathbf{w}), g_4(\mathbf{x}, \mathbf{y}, \mathbf{z}, \mathbf{w})), \\
g_1(\mathbf{x}, \mathbf{y}, \mathbf{z}, \mathbf{w}) &:= \mathbf{I}_n \mathbf{x} - 2\alpha \nabla_{\mathbf{x}} F(\mathbf{x}, \mathbf{y}, \mathbf{z}, \mathbf{w}) + \alpha \nabla_{\mathbf{z}} F(\mathbf{z}, \mathbf{w}, \mathbf{x}, \mathbf{y}), \\
g_2(\mathbf{x}, \mathbf{y}, \mathbf{z}, \mathbf{w}) &:= \mathbf{I}_m \mathbf{y} + 2\alpha \nabla_{\mathbf{y}} F(\mathbf{x}, \mathbf{y}, \mathbf{z}, \mathbf{w}) - \alpha \nabla_{\mathbf{w}} F(\mathbf{z}, \mathbf{w}, \mathbf{x}, \mathbf{y}), \\
g_3(\mathbf{x}, \mathbf{y}, \mathbf{z}, \mathbf{w}) &:= \mathbf{I}_n \mathbf{x}, \\
g_4(\mathbf{x}, \mathbf{y}, \mathbf{z}, \mathbf{w}) &:= \mathbf{I}_m \mathbf{y}.
\end{aligned}
\tag{7}
$$

It is not hard to check that $(\mathbf{x}_{t+1}, \mathbf{y}_{t+1}, \mathbf{x}_t, \mathbf{y}_t) = g(\mathbf{x}_t, \mathbf{y}_t, \mathbf{x}_{t-1}, \mathbf{y}_{t-1})$, so $g$ captures exactly the dynamics of OGDA (4). The idea behind the construction of the dynamical system above is common in the literature of ODEs (ordinal differential equations) where in order to solve (typically to understand the qualitative behavior) a higher order ODE, one approach is to express it as a linear system of ODEs.

## 3.2   Analyzing OGDA via system (7)

As in the case of GDA, we need to show the following key lemma in order to use the Center-stable manifold theorem.

**Lemma 3.1** (**OGDA is a local diffeomorphism**). *Let $f$ is real-valued $C^2$ and $\nabla f$ is Lipschitz with constant $L$ and $0 < \alpha < \frac{1}{L}$. Under the Assumption 1.7 we get that the update rule $g$ of the OGDA dynamics (7) is a local diffeomorphism.*

Again as in Section 2, we are able to prove the following measure zero argument using Lemma 3.1 and Center-Stable manifold theorem.

**Theorem 3.2** (**Measure zero for OGDA**). *Let $f$ be twice differentiable and $\nabla f$ is Lipschitz with constant $L$. Suppose that Assumption 1.7 holds and $0 < \alpha < \frac{1}{L}$. Let $g$ be the update rule of the OGDA dynamics (4), $(\mathbf{x}^*, \mathbf{y}^*, \mathbf{x}^*, \mathbf{y}^*)$ be a OGDA-unstable critical point and $W_{OGDA}(x^*, y^*, x^*, y^*)$ be its stable set, i.e.,*

$$W_{OGDA}(x^*, y^*, x^*, y^*) = \{(x_1, y_1, x_0, y_0) : \lim_k g^k(x_1, y_1, x_0, y_0) = (x^*, y^*, x^*, y^*)\}.$$

*It holds that $W_{OGDA}(x^*, y^*, \mathbf{x}^*, \mathbf{y}^*)$ is of Lebesgue measure zero. Moreover if $W_{OGDA}$ is union of the stable sets of all OGDA-unstable critical points, then $W_{OGDA}$ has also measure zero (namely the proof works for non-isolated critical points).*

The following corollary is immediate from Theorem 3.2.

**Corollary 3.3.** *Let $(\mathbf{x}^*, \mathbf{y}^*, \mathbf{x}^*, \mathbf{y}^*)$ be OGDA-unstable. Assume $\mu$ is a measure of the starting points $(\mathbf{x}_1, \mathbf{y}_1, \mathbf{x}_0, \mathbf{y}_0)$ and is absolutely continuous with respect to the Lebesgue measure on $\mathbb{R}^{2n+2m}$. Then it holds that*

$$Pr[\lim_t (\mathbf{x}_t, \mathbf{y}_t, \mathbf{x}_{t-1}, \mathbf{y}_{t-1}) = (\mathbf{x}^*, \mathbf{y}^*, \mathbf{x}^*, \mathbf{y}^*)] = 0.$$

## 3.3 Characterizing OGDA-stability

In this subsection we provide an analysis for the eigenvalues of the Jacobian matrix $J_{\text{OGDA}}$ of the update rule $g$ of the system (7) the equations of which can be found in the supplementary material. We begin by claiming that the set of GDA-stable critical points is a subset of the set of OGDA-critical points. We manage to show this by constructing a mapping between the eigenvalues of $J_{\text{GDA}}$ and $J_{\text{OGDA}}$.

**Lemma 3.4** (**GDA-stable are OGDA-stable**). *Let $f$ be twice differentiable and $\nabla f$ be $L$-Lipschitz. Assume that $0 < \alpha < \frac{1}{2L}$ and suppose $(\mathbf{x}^*, \mathbf{y}^*)$ is a critical point that is GDA-stable (i.e., stable according to dynamics (3)). The critical point $(\mathbf{x}^*, \mathbf{y}^*, \mathbf{x}^*, \mathbf{y}^*)$ is stable according to OGDA dynamics (4).*

We conclude the subsection with the following claim and a remark.

**Lemma 3.5.** *There are functions with critical points that are OGDA-stable but not GDA-stable.*

**Remark 3.6.** *We would like to note that some of our results (e.g., Lemma 3.1 and Theorem 3.2) are not applicable to a generic bilinear function $f(\mathbf{x}, \mathbf{y}) = \mathbf{x}^\top A \mathbf{y}$, since if $A$ is not a square matrix, the Hessian $\nabla^2 f$ is not invertible (they are applicable only when $A$ is square matrix and invertible).*

## 4 Examples and Experiments

In this section we provide two examples/experiments, one 2-dimensional (function $f : \mathbb{R}^2 \to \mathbb{R}, x, y \in \mathbb{R}$) and one higher dimensional ($f : \mathbb{R}^{10} \to \mathbb{R}, \mathbf{x}, \mathbf{y} \in \mathbb{R}^5$). The purpose of these experiments is to get better intuition about our findings. In the 2-dimensional example, we construct a function with local min-max, {GDA, OGDA}-unstable and {GDA, OGDA}-stable critical points. Moreover, we get 10000 random initializations from the domain $R = \{(x, y) : -5 \le x, y \le 5\}$ and we compute the probability to reach each critical point for both GDA and OGDA dynamics. In the higher dimensional experiment, we construct a polynomial function $p(\mathbf{x}, \mathbf{y})$ of degree 3 with coefficients sampled i.i.d from uniform distribution with support $[-1, 1]$ and then we plant a local min max. Under 10000 random initializations in $R$, we analyze the convergence properties of GDA and OGDA (as in the two dimensional case).

### 4.1 A 2D example

The function $f_1(x, y) = -\frac{1}{8}x^2 - \frac{1}{2}y^2 + \frac{6}{10}xy$ has the property that the critical point $(0, 0)$ is GDA-stable but not local min-max (see Lemma 2.5). Moreover, consider $f_2(x, y) = \frac{1}{2}x^2 + \frac{1}{2}y^2 + 4xy$. This function has the property that the critical point $(0, 0)$ is GDA-unstable and is easy to check that is not a local min-max. We construct the polynomial function $f(x, y) = f_1(x, y)(x-1)^2(y-1)^2 + f_2(x, y)x^2y^2$. Function $f$ has the property that around $(0, 0)$ behaves like $f_1$ and around $(1, 1)$ behaves like $f_2$. The GDA dynamics of $f$ can be seen in Figure 2. However more critical points are created. There are five critical points, i.e, $(0, 0), (0, 1), (1, 0), (1, 1), (0.3301, 0.3357)$ (in interval $R$, the last critical point is computed approximately). In Table 1 we observe that the critical point $(0, 0)$ is stable for OGDA but unstable for GDA (essentially OGDA has more attracting critical points). Moreover, our theorems of avoiding unstable fixed points are verified with this experiment. Note that there are some initial conditions that GDA and OGDA dynamics don't converge (3% and 9.8% respectively).

| Critical point | GDA-stable | OGDA-stable | Local min-max | value of $f$ | Prob. GDA converges | Prob. OGDA converges |
|---|---|---|---|---|---|---|
| $(0, 0)$ | NO | YES | NO | 0 | 0% | 25.8% |
| $(0, 1)$ | NO | NO | NO | 0 | 0% | 0% |
| $(1, 0)$ | YES | YES | YES | 0 | 78% | 35.4% |
| $(1, 1)$ | YES | YES | NO | 0 | 19% | 29% |
| $(0.3301, 0.3357)$ | NO | NO | NO | 0.109 | 0% | 0% |

Table 1: Summary of critical points of $f$.

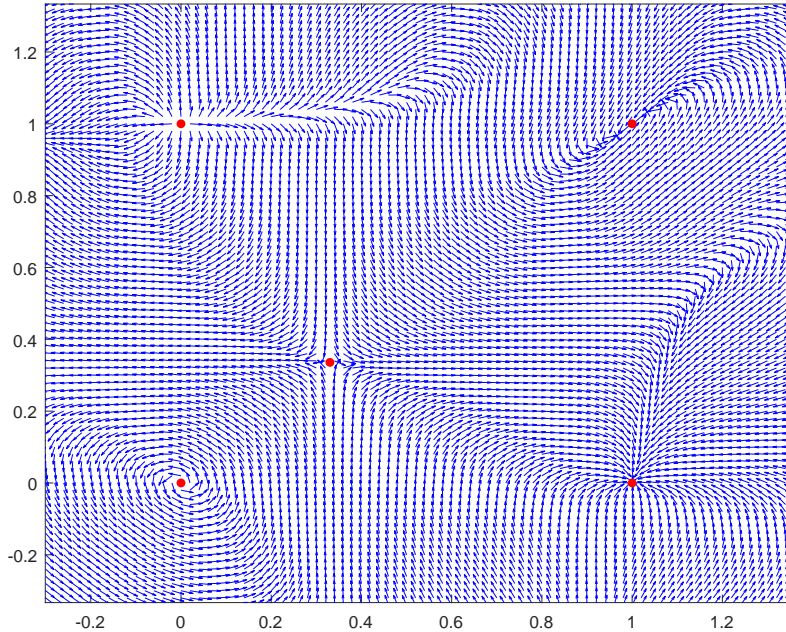

Figure 2: Construction of a function with points that are GDA-stable and local min-max, GDA-stable and not local min-max and GDA-unstable (and hence not local min-max). The arrows point towards the next step of the Gradient Descent/Ascent dynamics.

## 4.2 Higher dimensional

Let $f(\mathbf{x}, \mathbf{y}) := p(\mathbf{x}, \mathbf{y}) \cdot (\sum_{i=1}^{5} x_i^3 + y_i^3) + w(\mathbf{x}, \mathbf{y})$, where $p$ is the random 3-degree polynomial as mentioned above and $w(\mathbf{x}, \mathbf{y}) = \sum_{i=1}^{5} (x_i^2 - y_i^2)$. It is clear that $f$ locally at $(0, ..., 0)$ behaves like function $w$ (which has $\mathbf{0}$ as a local min-max critical point). We run for 10000 uniformly random points in $R$ and it turns out that 87% of initial points converge to $\mathbf{0}$ in OGDA as opposed to GDA which 79.3% fraction converged. This experiment indicates qualitative difference between the two methods, where the area of region of attraction in OGDA is a bit larger.

## 5 Conclusion

In this paper we made a step towards understanding first order methods which are used to solve min-max optimization problems, by analyzing the local behavior of GDA and OGDA dynamics around critical points. Our paper is an indication that important first order methods we analyze fail to converge to only local min-max solutions (standard concept in optimization literature). Whether or not local min-max solutions is a good concept is out of the scope of this paper[11]. Local min-max solutions might not be all equally good and some may be bad, which is really important in applications such as training GANs. Nevertheless, even for minimization problems, finding good local minima is a hard task that is not well understood in the literature (most first order methods guarantee convergence to some local minimum, without guarantees about its quality). A forteriori guaranteeing good solutions in a min-max problem is a harder proposition and an important open question.

**Acknowledgments**

Constantinos Daskalakis was supported by NSF awards CCF-1617730 and IIS-1741137, a Simons Investigator Award, a Google Faculty Research Award, and an MIT-IBM Watson AI Lab research grant. Ioannis Panageas was supported by SRG ISTD 2018 136. This work was done when Ioannis was a postdoctoral fellow at MIT.

## Footnotes

[1]We note that in their paper the dynamics is called Optimistic Mirror Descent, we changed the name because the dynamics is a modified gradient descent.

[2]Note that $\alpha > 0$ for the rest of this paper.

[3]We note that OGDA has some resemblance with Polyak's heavy ball method. However, one important difference is that OGDA has "negative momentum" while the heavy ball method has "positive momentum."

[4]A local diffeomorphism is a function that locally is invertible, smooth and its (local) inverse is also smooth.

[5]In optimization literature they are called *local saddles*.

[6]We also use $\mathbf{0}$ to denote the zero vector.

[7]Ball of radius $\delta$.

[8]If the critical points are isolated then they are countably many or finite.

[9]In optimization literature these critical points are also called local saddle points. If $U$ is the whole domain then we call it global min-max.

[10]See Figure 1.

[11]Even characterizing whether a local min-max solution is good or not is not an easy/clear task.

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
