[Supplementary Material · supplementary.pdf]

# *Supplementary material* to The Limit Points of (Optimistic) Gradient Descent in Min-Max Optimization

## A   Missing theorems and proofs

*Proof of Lemma 2.1.* Let $h(\mathbf{x}, \mathbf{y})$ be the update rule of the dynamics (3). It suffices to show that the Jacobian $J_{\text{GDA}}$ of $h$ is invertible and by the use of Inverse Function theorem, the claim follows. After straightforward calculations we get

$$J_{\text{GDA}} = \begin{pmatrix} \mathbf{I}_n - \alpha \nabla^2_{\mathbf{xx}} f & -\alpha \nabla^2_{\mathbf{xy}} f \\ \alpha \nabla^2_{\mathbf{yx}} f & \mathbf{I}_m + \alpha \nabla^2_{\mathbf{yy}} f \end{pmatrix}, \tag{8}$$

where the Hessian of $f$ is given by

$$\nabla^2 f = \begin{pmatrix} \nabla^2_{\mathbf{xx}} f & \nabla^2_{\mathbf{xy}} f \\ \nabla^2_{\mathbf{yx}} f & \nabla^2_{\mathbf{yy}} f \end{pmatrix}. \tag{9}$$

It suffices to show that the matrix below does not have an eigenvalue that is equal to $-1/\alpha$ (by just subtracting the identity matrix),

$$H_{\text{GDA}} = \begin{pmatrix} -\nabla^2_{\mathbf{xx}} f & -\nabla^2_{\mathbf{xy}} f \\ \nabla^2_{\mathbf{yx}} f & \nabla^2_{\mathbf{yy}} f \end{pmatrix}. \tag{10}$$

It is easy to see that

$$H_{\text{GDA}} = \begin{pmatrix} -\mathbf{I}_n & \mathbf{0}_{n \times m} \\ \mathbf{0}_{m \times n} & \mathbf{I}_m \end{pmatrix} \left( \nabla^2 f \right). \tag{11}$$

If the function $\nabla f$ is L-Lipschitz, it follows that $\left\| \nabla^2 f \right\|_2 \leq L$ (Lemma 6 in [6]). Therefore by equation (11) we have that $\rho(H_{\text{GDA}}) \leq \|H_{\text{GDA}}\|_2 \leq \left\| \nabla^2 f \right\|_2 \leq L < \frac{1}{\alpha}$. The claim follows.   □

*Proof of Lemma 2.4.* By definition of local min-max, it holds that $\nabla^2_{\mathbf{xx}} f$ is positive semi-definite and also $\nabla^2_{\mathbf{yy}} f$ is negative semi-definite. Hence the symmetric matrix below (matrix $H_{\text{GDA}}$ is given by equation (10))

$$\frac{1}{2} \left( H_{\text{GDA}} + H_{\text{GDA}}^\top \right) = \begin{pmatrix} -\nabla^2_{\mathbf{xx}} f & \mathbf{0}_{n \times m} \\ \mathbf{0}_{m \times n} & \nabla^2_{\mathbf{yy}} f \end{pmatrix}$$

is negative semi-definite. We use the Ky Fan inequality which states that the sequence (in decreasing order) of the eigenvalues of $\frac{1}{2}(H_{\text{GDA}} + H_{\text{GDA}}^\top)$ majorizes the real part of the sequence of the eigenvalues of $H_{\text{GDA}}$ (see [5], page 4). By assumption that $H_{\text{GDA}}$ has real eigenvalues we conclude that $\lambda_{\max}(H_{\text{GDA}}) \leq \frac{1}{2}\lambda_{\max}(H_{\text{GDA}} + H_{\text{GDA}}^\top) \leq 0$ since $H_{\text{GDA}} + H_{\text{GDA}}^\top$ is negative semi-definite. Therefore the spectrum of $I + \alpha H_{\text{GDA}}$ lies in $[-1, 1]$ (since also $\alpha < 1/L$), thus $(\mathbf{x}^*, \mathbf{y}^*)$ is GDA-stable.   □

*Proof of Lemma 2.6.* Let $f(x, y) = xy$. It is clear that critical point $(0, 0)$ is a local min-max point. Computing the Jacobian of the update rule of dynamics (3) at point $(0, 0)$ we get that

$$J_{\text{GDA}} = \begin{pmatrix} 1 & -\alpha \\ \alpha & 1 \end{pmatrix}, \tag{12}$$

For any $\alpha > 0$ we have that the eigenvalues of $J_{\text{GDA}}$ are $1 \pm \alpha i$,[1] so they have magnitude greater than 1 (and is clear that $H_{\text{GDA}}$ has complex eigenvalues). It is easy to see that $x_{t+1}^2 + y_{t+1}^2 = (1 + \alpha^2)(x_t^2 + y_t^2)$, i.e., inductively we have

$$x_t^2 + y_t^2 = (1 + \alpha^2)^t (x_0^2 + y_0^2),$$

hence GDA dynamics diverges. $\square$

*Proof of Lemma 2.7.* The proof follows the steps of the proof of Lemma 2.4. Similarly, using Ky Fan inequality we know that for any eigenvalue $\lambda$ of $H_{\text{GDA}}$ it holds that

$$\text{Re}(\lambda) \leq \frac{1}{2} \lambda_{\max}(H_{\text{GDA}} + H_{\text{GDA}}^\top) \leq 0.$$

Hence we conclude that $\text{Re}(\lambda) < 0$. Additionally, the corresponding eigenvalue of $J_{\text{GDA}}$ is $1 + \alpha\lambda$. By choosing $\alpha < \min_\lambda \{-\frac{\text{Re}(\lambda)}{|\lambda|^2}\}$, it is easy to see that $|1 + \alpha\lambda|^2 = 1 + \alpha\text{Re}(\lambda) + \alpha^2|\lambda|^2 < 1$ for all the eigenvalues $\lambda$ of $H_{\text{GDA}}$, hence the eigenvalues of $J_{\text{GDA}}$ have magnitude less than one. $\square$

*Proof of Lemma 3.1.* It suffices to show that the Jacobian of $g$, denoted by $J_{\text{OGDA}}$ is invertible and then by Inverse Function theorem the claim follows. After calculations the Jacobian boils down to the following (we set $F'(\mathbf{x}, \mathbf{y}, \mathbf{z}, \mathbf{w}) = F(\mathbf{z}, \mathbf{w}, \mathbf{x}, \mathbf{y})$):

$$J_{\text{OGDA}} = \begin{pmatrix} \mathbf{I}_n - 2\alpha\nabla_{\mathbf{xx}}^2 F & -2\alpha\nabla_{\mathbf{xy}}^2 F & \alpha\nabla_{\mathbf{zz}}^2 F' & \alpha\nabla_{\mathbf{zw}}^2 F' \\ 2\alpha\nabla_{\mathbf{yx}}^2 F & \mathbf{I}_m + 2\alpha\nabla_{\mathbf{yy}}^2 F & -\alpha\nabla_{\mathbf{wz}}^2 F' & -\alpha\nabla_{\mathbf{ww}}^2 F' \\ \mathbf{I}_n & \mathbf{0}_{n\times m} & \mathbf{0}_{n\times n} & \mathbf{0}_{n\times m} \\ \mathbf{0}_{m\times n} & \mathbf{I}_m & \mathbf{0}_{m\times n} & \mathbf{0}_{m\times m} \end{pmatrix}, \quad (13)$$

Observe that for $\alpha = 0$, the matrix $J_{\text{GDA}}$ is not invertible, as opposed to the case of GDA which is the identity matrix $\mathbf{I}_{n+m}$ and hence is invertible. It is easy to see that for $\alpha = 0$, then $g(\mathbf{x}, \mathbf{y}, \mathbf{z}, \mathbf{w}) = (\mathbf{x}, \mathbf{y}, \mathbf{x}, \mathbf{y})$, namely it is not even $1 - 1$ (not even locally).

The null space of $J_{\text{OGDA}}$ is the same as the null space of the following matrix $H_{\text{OGDA}}$ (after row and column operations)

$$H_{\text{OGDA}} = \begin{pmatrix} \mathbf{0}_{n\times n} & \mathbf{0}_{n\times m} & \alpha\nabla_{\mathbf{zz}}^2 F' & \alpha\nabla_{\mathbf{zw}}^2 F' \\ \mathbf{0}_{m\times n} & \mathbf{0}_{m\times m} & -\alpha\nabla_{\mathbf{wz}}^2 F' & -\alpha\nabla_{\mathbf{ww}}^2 F' \\ \mathbf{I}_n & \mathbf{0}_{n\times m} & \mathbf{0}_{n\times n} & \mathbf{0}_{n\times n} \\ \mathbf{0}_{m\times n} & \mathbf{I}_m & \mathbf{0}_{m\times n} & \mathbf{0}_{m\times m} \end{pmatrix}, \quad (14)$$

It is clear that under the assumption that the Hessian is invertible (see Assumption 1.7), we get that

$$\begin{pmatrix} \nabla_{\mathbf{zz}}^2 F' & \nabla_{\mathbf{zw}}^2 F' \\ -\nabla_{\mathbf{wz}}^2 F' & -\nabla_{\mathbf{ww}}^2 F' \end{pmatrix} \text{ is invertible} \quad (15)$$

and so is $H_{\text{OGDA}}$. $\square$

*Proof of Lemma 3.4.* A fixed point of the dynamics (4) is of the form $(\mathbf{x}, \mathbf{y}, \mathbf{x}, \mathbf{y})$ (see Remark 1.5). The Jacobian of the update rule $g$ becomes as follows:

$$J_{\text{OGDA}} = \begin{pmatrix} \mathbf{I}_n - 2\alpha\nabla_{\mathbf{xx}}^2 F & -2\alpha\nabla_{\mathbf{xy}}^2 F & \alpha\nabla_{\mathbf{xx}}^2 F & \alpha\nabla_{\mathbf{xy}}^2 F \\ 2\alpha\nabla_{\mathbf{yx}}^2 F & \mathbf{I}_m + 2\alpha\nabla_{\mathbf{yy}}^2 F & -\alpha\nabla_{\mathbf{yx}}^2 F & -\alpha\nabla_{\mathbf{yy}}^2 F \\ \mathbf{I}_n & \mathbf{0}_{n\times m} & \mathbf{0}_{n\times n} & \mathbf{0}_{n\times m} \\ \mathbf{0}_{m\times n} & \mathbf{I}_m & \mathbf{0}_{m\times n} & \mathbf{0}_{m\times m} \end{pmatrix}. \quad (16)$$

We would like to find a relation between the eigenvalues of matrix (16) and matrix (8) (relate the Jacobian of both dynamics GDA and OGDA). We start with the matrix

$$\lambda\mathbf{I}_{2m+2n} - J_{\text{OGDA}} = \begin{pmatrix} \lambda\mathbf{I}_n - \mathbf{I}_n + 2\alpha\nabla_{\mathbf{xx}}^2 F & 2\alpha\nabla_{\mathbf{xy}}^2 F & -\alpha\nabla_{\mathbf{xx}}^2 F & -\alpha\nabla_{\mathbf{xy}}^2 F \\ -2\alpha\nabla_{\mathbf{yx}}^2 F & \lambda\mathbf{I}_m - \mathbf{I}_m - 2\alpha\nabla_{\mathbf{yy}}^2 F & \alpha\nabla_{\mathbf{yx}}^2 F & \alpha\nabla_{\mathbf{yy}}^2 F \\ -\mathbf{I}_n & \mathbf{0}_{n\times m} & \lambda\mathbf{I}_n & \mathbf{0}_{n\times m} \\ \mathbf{0}_{m\times n} & -\mathbf{I}_m & \mathbf{0}_{m\times n} & \lambda\mathbf{I}_m \end{pmatrix}.$$

The absolute value of the determinant of a matrix remains invariant under row/column operations (add a multiple of a row/column to another row/column or exchange rows/columns). After such operations, the determinant of the matrix above has determinant in absolute value equal to (we assume that $\lambda \neq 0$)

$$\det \begin{pmatrix} \lambda \mathbf{I}_n - \mathbf{I}_n + (2 - 1/\lambda)\alpha \nabla^2_{\mathbf{xx}}F & (2 - 1/\lambda)\alpha \nabla^2_{\mathbf{xy}}F & -\alpha \nabla^2_{\mathbf{xx}}F & -\alpha \nabla^2_{\mathbf{xy}}F \\ (1/\lambda - 2)\alpha \nabla^2_{\mathbf{yx}}F & \lambda \mathbf{I}_m - \mathbf{I}_m + (1/\lambda - 2)\alpha \nabla^2_{\mathbf{yy}}F & \alpha \nabla^2_{\mathbf{yx}}F & \alpha \nabla^2_{\mathbf{yy}}F \\ \mathbf{0}_{n \times n} & \mathbf{0}_{n \times m} & \lambda \mathbf{I}_n & \mathbf{0}_{n \times m} \\ \mathbf{0}_{m \times n} & \mathbf{0}_{m \times m} & \mathbf{0}_{m \times n} & \lambda \mathbf{I}_m \end{pmatrix}.$$

The determinant above is equal to $\lambda^{m+n}p(\lambda)$, where

$$p(\lambda) = \det \begin{pmatrix} \lambda \mathbf{I}_n - \mathbf{I}_n + (2 - 1/\lambda)\alpha \nabla^2_{\mathbf{xx}}F & (2 - 1/\lambda)\alpha \nabla^2_{\mathbf{xy}}F \\ (1/\lambda - 2)\alpha \nabla^2_{\mathbf{yx}}F & \lambda \mathbf{I}_m - \mathbf{I}_m + (1/\lambda - 2)\alpha \nabla^2_{\mathbf{yy}}F \end{pmatrix}.$$

It is clear that $\lambda = \frac{1}{2}$ is not an eigenvalue of $J_{\text{OGDA}}$. Let $q_{\text{GDA}}(\lambda)$ be the characteristic polynomial of $J_{\text{GDA}}$ (8, Jacobian of GDA dynamics at $(\mathbf{x}, \mathbf{y})$). The characteristic polynomial $q_{\text{OGDA}}$ of $J_{\text{OGDA}}$ ends up being equal to

$$\det \begin{pmatrix} \lambda^2 \mathbf{I}_n - \lambda \mathbf{I}_n + (2\lambda - 1)\alpha \nabla^2_{\mathbf{xx}}F & (2\lambda - 1)\alpha \nabla^2_{\mathbf{xy}}F \\ -(2\lambda - 1)\alpha \nabla^2_{\mathbf{yx}}F & \lambda^2 \mathbf{I}_m - \lambda \mathbf{I}_m - (2\lambda - 1)\alpha \nabla^2_{\mathbf{yy}}F \end{pmatrix}.$$

Therefore

$$q_{\text{OGDA}}(\lambda) = (2\lambda - 1)^{n+m} q_{\text{GDA}}\left(\frac{\lambda^2 + \lambda - 1}{2\lambda - 1}\right). \tag{17}$$

Let $r$ be an eigenvalue of matrix $H_{\text{GDA}}$ (10), i.e., $r + 1$ is an eigenvalue of $J_{\text{GDA}}$. From relation (17) it turns out that the roots of the polynomial

$$\lambda^2 - \lambda(1 + 2r) + r = 0, \tag{18}$$

are eigenvalues of the matrix $J_{\text{OGDA}}$. For $\alpha < \frac{1}{2L}$ it holds that $|r| < \frac{1}{2}$ and it turns out that all the roots of the quadratic equation (18) have magnitude at most one (see Mathematica code in Section A.1 for a proof of the inequality). $\qquad \square$

*Proof of Lemma 3.5.* The easiest example is $f(x, y) = xy$. It is clear that the Jacobian of GDA dynamics (3) is given by

$$J = \begin{pmatrix} 1 & -\alpha \\ \alpha & 1 \end{pmatrix}, \tag{19}$$

which has eigenvalues $1 \pm \alpha i$ (magnitude greater than one) and hence the critical point $(0, 0)$ is GDA-unstable. However, the Jacobian of OGDA dynamics (4) is given by

$$J_{\text{OGDA}} = \begin{pmatrix} 1 & -2\alpha & 0 & \alpha \\ 2\alpha & 1 & -\alpha & 0 \\ 1 & 0 & 0 & 0 \\ 0 & 1 & 0 & 0 \end{pmatrix}, \tag{20}$$

which has the four eigenvalues $\frac{1}{2}(1 \pm \sqrt{1 - 8\alpha^2 \pm 4\sqrt{4\alpha^4 - \alpha^2}})$. For $0 < \alpha < 1/2$ all the four eigenvalues have magnitude less than or equal to 1, hence $(0, 0)$ is OGDA-stable (see mathematica code A.2 for the inequality claim). Another example which is not bilinear (Assumption 1.7 is satisfied) is the function $\frac{1}{2}x^2 + \frac{1}{2}y^2 + 4xy$ (this is used in the example section). $\qquad \square$

**Theorem A.1** (Center-stable manifold theorem, III.7 [7])**.** *Let $x^*$ be a fixed point for the $C^r$ local diffeomorphism $g : \mathcal{X} \to \mathcal{X}$. Suppose that $E = E_s \oplus E_u$, where $E_s$ is the span of the eigenvectors corresponding to eigenvalues of magnitude less than or equal to one of $Dg(x^*)$, and $E_u$ is the span of the eigenvectors corresponding to eigenvalues of magnitude greater than one of $Dg(x^*)$[2]. Then there exists a $C^r$ embedded disk $W^{cs}_{loc}$ of dimension $\dim(E^s)$ that is tangent to $E_s$ at $x^*$ called the local stable center manifold. Moreover, there exists a neighborhood $B$ of $x^*$, such that $g(W^{cs}_{loc}) \cap B \subset W^{cs}_{loc}$, and $\cap_{k=0}^{\infty} g^{-k}(B) \subset W^{cs}_{loc}$.*

*Proof of Theorem 2.2 and Theorem 3.2.* It follows the general line of the papers [2, 4, 3, 6, 1]. We assume that the update rule of GDA, OGDA dynamics is a diffeomorphism (as proved in Lemmas 2.1 and 3.1). The proof is generic and has appeared in [2]. Let $A$ be the set of unstable critical points $x^*$ of a dynamical system with update rule a function $g : \mathcal{X} \to \mathcal{X}$ (in $C^2$). For each $x^* \in A$, there is an associated open neighborhood $B_{x^*}$ promised by the Stable Manifold Theorem A.1. $\cup_{x^* \in A} B_{x^*}$ forms an open cover, and since $\mathcal{X}$ is second-countable we can extract a countable subcover, so that $\cup_{x^* \in A} B_{x^*} = \cup_{i=1}^\infty B_{x_i^*}$.

Define $W = \{x_0 : \lim_k x_k \in A\}$ (stable set of $A$). Fix a point $x_0 \in W$. Since $x_k \to x^* \in A$, then for some non-negative integer $T$ and all $t \geq T$, $g^t(x_0) \in \cup_{x^* \in A} B_{x^*}$. Since we have a countable sub-cover, $g^t(x_0) \in B_{x_i^*}$ for some $x_i^* \in A$ and all $t \geq T$. This implies that $g^t(x_0) \in \cap_{k=0}^\infty g^{-k}(B_{x_i^*})$ for all $t \geq T$. By Theorem A.1, $S_i \triangleq \cap_{k=0}^\infty g^{-k}(B_{x_i^*})$ is a subset of the local center stable manifold which has co-dimension at least one, and $S_i$ is thus measure zero.

Finally, $g^T(x_0) \in S_i$ implies that $x_0 \in g^{-T}(S_i)$. Since $T$ is unknown we union over all non-negative integers, to obtain $x_0 \in \cup_{j=0}^\infty g^{-j}(S_i)$. Since $x_0$ was arbitrary, we have shown that $W \subset \cup_{i=1}^\infty \cup_{j=0}^\infty g^{-j}(S_i)$. Using Lemma 1 of page 5 in [2] and that countable union of measure zero sets is measure zero, $W$ has measure zero.

$\square$

## A.1 Mathematica code for proving claim in Lemma 3.4

```
Reduce[Norm[r] < 1/2 && Norm[1 + r] < 1
&& (Norm[r + 1/2 - 1/2*Sqrt[4 r^2 + 1]] > 1
|| Norm[r + 1/2 + 1/2*Sqrt[4 r^2 + 1]] > 1), r, Complexes]

False
```

## A.2 Mathematica code for proving claim in Lemma 3.5

```
Reduce[Abs[1/2 (1 + Sqrt[1 - 8 x^2 + 4 Sqrt[-x^2 + 4 x^4]])] > 1 &&
  0 < x < 1/2]

False

Reduce[Abs[1/2 (1 - Sqrt[1 - 8 x^2 - 4 Sqrt[-x^2 + 4 x^4]])] > 1 &&
  0 < x < 1/2]

False

Reduce[Abs[1/2 (1 + Sqrt[1 - 8 x^2 - 4 Sqrt[-x^2 + 4 x^4]])] > 1 &&
  0 < x < 1/2]

False

Reduce[Abs[1/2 (1 - Sqrt[1 - 8 x^2 + 4 Sqrt[-x^2 + 4 x^4]])] > 1 &&
  0 < x < 1/2]

False
```

## Footnotes

[1] We denote $i := \sqrt{-1}$.

[2]Jacobian of function $g$.