[Reviews · NeurIPS 2018]

Reviewer 1



Review: This paper studies the problem of understanding the limit points of Gradient Descent Ascent (GDA) and an optimistic version of the same (OGDA) for min-max optimization. There are two main results from the paper state that (under certain assumptions) a) The set of initial vectors from which GDA/OGDA converge to unstable fixed points are of Lebesgue measure zero b) The set of local min-max (saddle) points and the stable fixed points of GDA/OGDA satisfy the following (strict) inclusion: Local min-max \subset GDA-stable \subset OGDA-stable. A key technical contribution relates to the construction of the OGDA dynamical system and its analysis in order to understand the corresponding limit points. 1. While the paper makes an initial attempt at understanding basic first order methods and characterizing their "convergent" fixed points, as pointed out in the paper and also observed experimentally, there are several initial starting configurations for the OGD/OGDA dynamics fail to converge. In cases where they fail to converge, is it the case that the dynamics cycle through the unstable fixed points? While Corollary 2.3 and 3.3 shows that unstable points cannot be converged to, the paper should indeed make it clear if such cyclical/non-convergent behavior is still a possibility. 2. In several proofs in the appendix, the paper cites lemmas from other papers for completing the proof. In order to make the paper self-sufficient, I would recommend restating those lemmas in the appendix for the ease of the reader. Overall, this seems to be a well written paper which provides the first characterization of limit points of basic first order methods and raises several interesting questions which are important in understanding the problem of general min-max optimization.

Reviewer 2



The paper studies stability of gradient and optimistic gradient dynamics for C^2 saddle-point problems. The main contribution of the paper can be summarized in two results (stated in the inclusion following line 83): - local saddles are stable for GDA (under Assumption 8.1) - stable equilibria of GDA are also stable for OGDA. (note that results on unstable critical points were previously known). Quality: The results are interesting, and the paper is well written. There are some typos in the proofs, but I believe these are omissions that can be corrected, rather than major flaws. Significance: I would love to see further discussion of the consequences of this result, and its relevance to the NIPS community, both theoreticians and practitioners. For example, do these results suggest that GDA should be preferred to OGDA (since the latter has a larger equilibrium set)? Does the analysis extend to time-varying step sizes? In the concluding remarks, the authors mention that some saddle points (local min max) can be better than others. Can the authors expand on how the quality of an equilibrium can be measured? Originality: It takes some effort to tease out the contributions of the paper from existing results. Existing results should be cited but not part of the main paper (they can potentially be moved to the appendix) in favor of an expanded discussion of Theorem 1.10 and its consequences. - Theorems 1.11, 2.2 and 3.2 (that a dynamical system a.s. does not converge to an unstable equilibrium when the Hessian is invertible) are known. Note that the particular form of the dynamics is irrelevant to the result, only the fact that the spectral radius of the Jacobian at the equilibrium is greater than one. I don't think one can claim this is a contribution of this paper. - Similarly, Corollary 2.3 and 3.3 are immediate consequences of these known results, and are stated without discussion. I do not see what they bring to the picture. - The literature review needs to be expanded. Similar techniques have been used to prove stability of gradient algorithms (e.g. [1]), and for GAN training [2], which seems particularly relevant, since they use similar techniques (analyzing the Jacobian at the equilibrium, albeit for continuous time dynamics) and this paper is motivated by GAN training. Typos and minor suggestions: - Line 18: "compact and concave subsets": do you mean convex? - In the analysis of OGDA dynamics, what is the advantage of introducing the functions F and F'? One can simply express the dynamics (and the proof of Lemma 3.1 in the appendix) in terms of f. Defining F, F' introduces unnecessary additional notation. - Second experiment (4.2): how is this experiment related to the results of the paper? - Statement of Theorem B.1: what is E? What is Dg(x^*), is this the Jacobian? Please use notation consistent with the rest of the paper. - Please define "C^r embedded disk" in Theorem B.1. - In the proof of Theorem 2.2 and 3.2, you can extract a finite sub-cover simply by the compactness assumption. Why mention second-countability? - In the proof of Lemma 2.4: I + a H should be I + \alpha H. The proof only provides an upper bound on the eigenvalues of H. Note that this is not sufficient, a lower bound is also needed to conclude. - Proof of Lemma 2.7: Incomplete sentence "Hence we conclude that Re(\lambda)". There is also a missing factor 2 in the expression of the eigenvalue magnitude. [1] L. Lessard, B. Recht and A. Packard. Analysis and Design of Optimization Algorithms via Integral Quadratic Constraints. SIAM Journal on Optimization - 26(1): 57-95 [2] V. Nagarajan and J. Zico Kolter, Gradient descent GAN optimization is locally stable. NIPS 2017. ======== post rebuttal Thank you for your responses, and for the corrections to the proofs. Regarding Theorem 2.2, 3.2 and Corollaries 2.3 and 3.3: I agree that proving the invertibility of the Jacobian for these particular dynamics is new, but the rest of the argument is identical to previous results, and this minor contribution hardly justifies stating two lemmas, two theorems and two corollaries, especially when much of this space can be used to improve the discussion of the new results. I strongly encourage the authors to move these to the supplement, and clarify that these mostly follow existing arguments.

Reviewer 3



Update: I have the read the authors' response. As the authors seem to insist on maintaining the current (and in my opinion misleading) framing of the results, I reduced my score by 1. Regarding my comment (1), *any iterative search method avoids its unstable points* - this is how unstable points are defined! Thus Theorems 2.2 and 3.3 are not "important global properties", but merely statements that definitions are self-consistent. --- The authors study the limiting behaviour of two first-order methods for zero-sum games, which they refer to as gradient descent/ascent (GDA) and optimistic gradient descent/ascent (OGDA). Assuming favorable function structure, the authors show that local min-max points (i.e. local saddles/Nash equilibria) are attractors for both GDA and OGDA, and that attractors for GDA are also attractors for OGDA. The authors also demonstrate the converses are false: they construct functions with attractors for OGDA which are not attractors for GDA, and attractors for GDA which are not local min-max points. These functions also exhibit initializations for which GDA and OGDA do not converge at all. The authors also apply the stable manifold argument to show that the set of initializations of GDA and OGDA that converge to non-attractors has measure zero. My opinion regarding this paper is divided. On the one hand, the paper reveals the fact that, even under very favorable conditions, GDA and OGDA can converge to points that are not even local solution to the game they are applied on - this is an important observation and cause for concern considering the recent popularity of adversarial training. On the other hand, the paper buries the said important observation behind unenlightening formalisms (see more details below). In reading the paper I got the feeling that the authors set out to show theoretically that GDA/OGDA are "good" in some sense, and are trying to downplay the fact that their results indicate the opposite. Moreover, the writing in the paper is unpolished, with many typos and somewhat disorganized presentation. Weighting the pros and cons above, I lean slightly towards accepting the paper, hoping the authors will improve the presentation in the final version of the paper, and frontload the more novel part of their results. Below are more detailed comments. 1) I do not see the point of the "stable manifold" results in this paper (i.e. Theorems 2.2 and 3.3). In smooth minimization problems, such results are important since they rigorously show convergence to the natural object of interest (local minima). Here, your results rule out convergence to the natural object of interest (local min-max points), and the stable manifold arguments reduce to near-tautologies, as your definition of "(O)GDA-unstable critical point" is tailored to make the stable manifold analysis go through. 2) The results of section 2 are essentially folklore and to some extent overlap with published literature - cf. "Characterization and Computation of Local Nash Equilibria in Continuous Games" by Ratliff et al. (2013). 3) As GANs appear to be the main motivators of this papers, you need to comment on all the assumptions you make that do not hold for GANs. Namely, noiseless gradients, invertible Hessian, and globally Lipschitz gradient are completely unrealistic. I am not sure how realistic Assumption 1.8 is - you should comment on the matter. 4) There seems to be some confusion regarding nomenclature of OGDA. To the best of my understanding this algorithm was first proposed recently Daskalakis et al., who referred to it as "optimistic mirror descent" (OMD). However, a different algorithm called "optimistic mirror descent" was proposed by Rakhlin and Sridharan in 2013. Since the "new" OMD is not a special case of the old one and does not enjoy its theoretical guarantees, I find its name quite inappropriate, and I am happy you do not adopt it. However, when reviewing the literature you should mention past names for the method, and explain why you choose to use a different name.